# Structure, Mechanical and Magnetic Properties of Selective Laser Melted Fe-Si-B Alloy

**DOI:** 10.3390/ma15124121

**Published:** 2022-06-09

**Authors:** Vadim Sufiiarov, Danil Erutin, Artem Kantyukov, Evgenii Borisov, Anatoly Popovich, Denis Nazarov

**Affiliations:** 1Insitiute of Machinery, Materials and Transport, Peter the Great St. Petersburg Polytechnic University, Polytechnicheskaya, 29, 195251 St. Petersburg, Russia; erutin@inbox.ru (D.E.); kantyukov.artem@mail.ru (A.K.); evgenii.borisov@icloud.com (E.B.); director@immet.spbstu.ru (A.P.); 2Research Centre “Innovative Technologies of Composite Nanomaterials”, St. Petersburg State University, Universitetskaya Nab, 7/9, 199034 St. Petersburg, Russia; dennazar1@yandex.ru

**Keywords:** selective laser melting, soft-magnetic alloy, FeSiB, magnetic properties, additive manufacturing

## Abstract

Original 1CP powder was studied and it was founded that powder material partially consists of the amorphous phase, in which crystallization begins at 450 °C and ends at 575 °C. Selective laser melting parameters were investigated through the track study, and more suitable ones were found: laser power *P* = 90, 120 W; scanning speed *V* = 1200 mm/s. Crack-free columnar elements were obtained. The sample obtained with *P* = 90 W, contains a small amount of amorphous phase. X-ray diffraction of samples shows the presence of α-Fe(Si) and Fe_2_B. SEM-image analysis shows the presence of ordered Fe_3_Si in both samples. Annealed samples show 40% less microhardness; an annealed sample containing amorphous phase shows higher soft-magnetic properties: 2.5% higher saturation magnetization, 35% higher residual magnetization and 30% higher rectangularity coefficient.

## 1. Introduction

Selective laser melting (SLM) technology is an additive manufacturing process by layer-by-layer melting of a 20–60 µm thick powder layer of material using a laser [1,2]. One of the most attractive applications of this technology is the production of bulk amorphous alloys.

Amorphous materials are a type of solid materials in which there is no long-range order in the arrangement of atoms [3]. This state of the material is achieved at high cooling rates from the liquid state due to the fixation of atoms in the positions in which they were in the melt. An amorphous metallic material (metallic glass) does not have a crystalline lattice; therefore, its atomic structure lacks the crystalline defects that cause anisotropy of its properties. Metallic glasses based on ferromagnetic alloys exhibit better soft-magnetic properties than crystalline ferromagnetics: lower coercive force values, higher values of saturation magnetization, higher magnetic permeability and electrical resistivity. Due to the higher level of soft magnetic properties of metallic ferromagnetic glasses, their use as a magnetic core material of an electromagnetic device allows for increasing its efficiency by significantly reducing the magnetic field energy losses for remagnetization and eddy currents [4].

High magnetic properties are achieved with amorphous phases, and materials consisting of nanocrystalline inclusions evenly distributed in the amorphous matrix are also known. Currently, amorphous and nanocrystalline materials produced by additive manufacturing techniques are being investigated by many research groups worldwide [1,4,5,6,7,8,9,10,11,12,13,14,15,16,17,18,19,20]. The problem of obtaining samples with amorphous phase from iron-based soft-magnetic alloys using SLM has been solved with varying degrees of success by selecting process parameters [1,4,11,14,18,19] and by developing and applying specially designed materials and laser beam scanning strategies [4,10,11,19]. Obtaining a material with a minimum degree of crystallinity involves the suppression of crystallization both when cooling the molten metal and when absorbing the heat of the melt by the already solidified part of the product. This task requires researchers to thoroughly understand the impact mechanisms on the resulting material, which makes investigation of the influence of process parameters on the characteristics of the resulting material a priority step towards its solution.

The aim of this work was to investigate the effect of the selective laser melting process parameters (laser power *P*, hatch distance *h*, offset *m*) on macrostructure and microstructure, phase composition, magnetic and mechanical properties of 1CP magnetic alloy samples and to investigate the effect of thermal treatment of samples on their magnetic and mechanical properties.

## 2. Materials and Methods

The flowability of the powder was determined using ISO 4490 “Determination of flow rate by means of a calibrated funnel (Hall flowmeter)”. Apparent density measurements were made by pouring the powder into a funnel from which it flowed into a 25 cm^3^ cup. After filling the cup, the funnel was moved away and excess powder was smoothed out with a trowel. Apparent density was determined by weighing the powder in the cup in grams and dividing by 25 cm^3^. The skeletal density of the powder was determined in accordance with GOST 22662-77.

The particle size distribution of the powder was determined by laser diffraction on the Analysette 22 NanoTec plus (Fritsch, Germany) with a total measuring range of 0.01–2000 µm. The microstructure of the powder and the obtained samples were studied using a Tescan Mira3 LMU scanning electron microscope (SEM). The etching of the samples was carried out in a 10% nitric acid solution in isopropyl alcohol. The fine structure of the powder was investigated using a Carl Zeiss Libra 200FE transmission electron microscope (TEM) with an energy Ω filter and an operating accelerating voltage of 200 kV. An HAADF detector (STEM mode) and a CCD camera (TEM mode) were used to obtain images. Electron diffraction patterns were measured using an aperture diameter of 1000 nm and a camera length of 450 mm. For the measurement of chemical composition, electron energy loss spectroscopy (EELS) in TEM was used and theoxygen content was measured by infrared absorption and thermal conductivity analysis on LECO TC-500 (LECO Corporation, St. Joseph, MI, USA).

Temperatures of phase transitions were studied using differential scanning calorimeter (DSC) Q2000 (TA Instruments, New Castle, DE, USA) equipped with an automatic sampler, RCS90 cooling system and T-zero baseline alignment technology. Samples were heated in an argon flow to a temperature of 1000 °C at a heating rate of 20 °C/min followed by second heating of the cooled samples to the same temperature. The phase composition was analyzed with a Bruker D8 Advance X-ray diffractometer (XRD) using Cu Kα (l 1/4 1.5418 Å) irradiation.

The magnetic hysteresis loops for the samples were measured by Lake Shore 7410 vibration sample magnetometer (VSM) (Lake Shore Cryotronics, Westerville, OH, USA) at room temperature (22 °C) and under applied different magnetic fields from −18,000 to +18,000 Oe. Magnetic measurements were carried out on 2 sets for each type of sample.

The hardness of the samples was determined using a Buehler Micromet 5103 micro-hardness tester using the Vickers method at 3 N. To determine the mean value, 5 tests were performed.

Samples were manufactured using an SLM280HL (SLM Solutions GmbH, Lübeck, Germany) selective laser melting system equipped with YLR-Laser (wavelength of 1070 nm and focus size about 80 μm) under a nitrogen atmosphere.

## 3. Results and Discussion

### 3.1. Powder Material

The first stage of the research was to investigate the morphology, phase composition, physical, technological and magnetic properties of the initial 1CP powder, consisting of iron and alloying elements: boron, carbon and silicon [21]. The chemical composition of the initial powder is presented in Table 1. The technology of the 1CP powder manufacturing was gas atomization [22].

The SEM image in Figure 1 shows that the powder material is spherical and rounded particles.

The results of the particle size distribution of the powder are shown in Table 2. The initial powder particle size is Gaussian distributed with a mean value of 41.8 μm. This is a typical range for use in selective laser melting [18,23].

The results of the investigation of the physical and technological properties of the powder are shown in Table 3. The ability to flow freely through the Hall funnel indicates the possibility of good powder spreading during the formation of thin powder layers in the selective laser melting process. The apparent density is 56.8% of the skeletal density, which indicates an acceptable packing density formed by this powder during the formation of the powder layer.

The X-ray diffraction pattern of the powder sample is shown in Figure 2. The following phases are present in the sample: solid solution α-Fe(Si) and iron boride Fe_2_B.

Figure 3 shows the DSC results of the powder material presented by two curves: the red curve for the primary heating of the original material and the blue one for the secondary heating of the material (cooled down after primary heating). The primary heating curve shows peaks indicating a phase transformation during heating. This process occurs for 1CP alloy powder in the temperature range from 450 to 575 °C. The absence of the secondary heating peaks shows the accordance of the peaks to the crystallization process. Based on the DSC data it could be concluded that heating above 450 °C would lead to the beginning of crystallization processes of the amorphous phase in case of the presence of the amorphous phase in the samples during heat treatment of samples obtained from this powder. According to this data, it was decided to use annealing heat treatment of samples at 440 °C. This annealing temperature corresponds to the recommended temperature for amorphous ribbons from 1CP [21] and the heat treatment mode used in further study: heating at a rate of 10 °C/min to 440 °C, holding for 30 min, cooling outside the furnace. The halo, which is not clearly visible in the diffraction pattern (Figure 2), is partially visible in the region of 2Θ = 42–47. The absence of an obvious halo can be explained by the small volume content of the amorphous phase in the powder material.

Figure 4 shows the results of investigation microstructure of 1CP powder by transmission electron microscopy.

Two phases are present in the studied powder, one of which has a crystalline structure as evidenced by electron diffraction (Figure 4d) and the other has an amorphous structure as evidenced by electron diffraction in Figure 4c. The amorphous phase is present both as separate areas (upper part of Figure 4a) and as areas distributed around the crystalline phase (Figure 4b and lower part of Figure 4a).

The results of the study on the magnetic properties of the powder are shown in Table 4. The hysteresis loop of the powder is shown in Figure 5. 1CP can be considered as a soft magnet with a relatively high coercive force and a low residual magnetization, but a huge saturation magnetization.

### 3.2. Single Track Study

In order to determine the range of applicability parameters for the selective laser melting process, a series of single tracks were melted on a 1CP substrate using different values of laser power *P* and scanning speed *V*, which were selected after the preliminary tests have been made with various values of laser power and scanning speed and provided continuous tracks.

The modes used for single-track series are presented in Table 5. The linear energy density is calculated as the ratio of a laser beam power to a scanning speed.

The SEM images of the tracks presented in Figure 6 show that there are transverse cracks repeated at distances greater than or close to 200–300 µm. A similar pattern is observed for all the tracks. Therefore, it was decided not to use values of one pass laser length exceeding 200 µm in the next experiments.

Figure 7 shows SEM images of the structure of the melted tracks in a cross-sectional view. The geometric characteristics of the resulting tracks are shown in Table 6. Track 1 has acceptable geometrical characteristics, but there is a pore in its cross-section and a crack at the border with the substrate. The linear energy density of the mode of this track is 75 J/m, as well as of track 4, which has good geometrical characteristics and has no visible defects, but the scanning speed used in the growth of the first track was too low for the used power, which led to the formation of defects. Tracks 2 and 3 were formed at lower linear energy densities (60 J/m and 50 J/m), which were insufficient to make a track with acceptable deposit height.

Tracks 4 and 5 have good geometrical characteristics (sufficient height of the deposited metal and penetration depth) and no visible defects, so modes 4 and 5 are used in the next experiments.

Thus, it was decided to use a melt track length not exceeding 200 µm, with values of *P* = 90 W, *P* = 120 W and *V* = 1200 mm/s.

### 3.3. Selective Laser Melting of Samples Investigation

As part of the study, eight rectangular samples were manufactured. Samples have been successively made in a nitrogen atmosphere. The plane orthogonal to the height of the cube was divided into cells, each containing two cross-sections of columnar elements, the distance between the centers of which corresponds to the hatch distance parameter h, with the distance between cells corresponding to the offset parameter m. The length of one pass of the laser beam also corresponds to the parameter *h*. The building scheme is presented in Figure 8.

The image of manufactured samples is shown in Figure 9. The build modes are presented in Table 7, the scanning speed *V* and the thickness of the powder layer *t* were fixed *V* = 1200 mm/s, *t* = 50 µm.

The samples manufactured at *P* = 90 W (Figure 10, 1–4) are less dense than those made at *P* = 120 W (Figure 10, 5–8). Increasing laser power allows the formation of larger structural elements due to the melting of a larger volume of initial powder, which leads to the formation of a denser structure. The increasing value of the offset (Figure 10, 1–4; 5–8) is accompanied by a decrease in the density of samples due to a violation of its structural unity caused by the separation of the columnar elements from each other. The hatch distance parameter h determines the presence of a merger of a pair of columnar elements into a single element: the samples obtained at *h* = 100 μm (Figure 10, 1–3; 5–7) are characterized by united elements, in contrast to the samples obtained at *h* = 200 μm (Figure 10, 4; 8).

Cross-sectional specimens were prepared for selected columnar elements of samples 4 and 8 (for these samples only the separation of single elements was possible) for examination with a scanning electron microscope. SEM images of the microstructure of the elements are shown in Figure 11 and Figure 12.

The structure of element 8 is characterized by the shape of the layer expressed by the presence of an arc section on the boundary line of each layer. This phenomenon is associated with increased laser power *P*, the value of which for sample 8 was 120 W. In this case, the change in the shape of the layer is associated with deeper penetration of laser irradiation for a separate section of the layer and uneven distribution of thermal energy over the contact spot of the laser with the metal.

The phase composition was investigated by X-ray diffraction analysis. The X-ray diffraction patterns of the samples are shown in Figure 13.

Based on X-ray diffraction analysis it was found that the following phases are present in the sample: α-Fe(Si) solid solution and Fe_2_B iron boride. The third phase present in the microstructure images of the samples can be identified as an ordered Fe_3_Si solid solution. The morphology of the etched cavities is similar to the crystal morphology of this phase [24]. The α-Fe(Si) solid solution has a similar crystallographic structure to the ordered Fe_3_Si solid solution, due to which the X-ray diffraction analysis may not allow the detection of the reflexes of this phase if the α-Fe(Si) structure prevails [24]. Therefore, researchers [19,23] during the X-ray diffraction analysis of samples of Fe-Si-B alloy obtained by selective laser melting noted the Fe_3_Si phase together with α-Fe(Si) on the peaks corresponding to α-Fe(Si).

The obtained DSC curves (Figure 14) indicate the almost complete absence of crystallization processes during the heating of the samples. However, the curve of primary heating of sample 4 is characterized by the presence of small peaks, and their absence during secondary heating (which cannot be said for the curve of sample 8), indicating the presence of a small amount of amorphous phase in sample 4.

Onset crystallization temperatures and enthalpy of the process are presented in Table 8. TEM electron diffraction data presented in Figure 15 proves the presence of the amorphous phase of sample 4.

Mechanical and magnetic properties were investigated for the initial and annealed samples. The purpose of annealing was to decrease the level of internal stresses in samples and investigate the effect of it for properties as defined above. The hardness data of the samples (Table 9) indicate that the hardness of sample 4 is slightly higher than that of sample 8. The difference between the mean hardness values is within the standard deviation of the samples (σ_4_ = 155, σ_8_ = 92). Hence, the laser power has no effect on the samples of this material in the investigated power range. The authors [1] investigated samples of a similar composition alloy and obtained hardness values close to those presented in this study. The annealed samples show an approximately 40% reduction values of hardness.

The study of the magnetic properties was carried out for samples 4 and 8. The magnetization curves of the samples are shown in Figure 16. The magnetization curves of the samples after heat treatment are shown in Figure 17. The main parameters of magnetic measurements are summarized in Table 8. The coercivity of the measured samples does not differ significantly from each other. At the same time, there is a difference in the shape of the hysteresis loop (sample 8 achieves a saturation at slightly lower values of field) and the values of residual magnetization. A comparison of the obtained results with the data for amorphous ribbons obtained by melt spinning technology for 1CP alloy [21] shows that the coercivity is much higher and the coefficient of rectangularity is lower for samples made by SLM.

The change of coercivity of annealed samples is within the margin of error. Sample 4 showed higher values of saturation magnetization (2.5% higher), residual magnetization (35% higher) and rectangularity coefficient (30% higher) after annealing. At the same time, sample 8 after heat treatment shows almost the same values of magnetic parameters as before heat treatment. The changing of magnetic properties for sample 4 is possibly related to a relaxation of internal stresses and the presence of a small amount of amorphous phase, magnetic properties changing of which after annealing is stronger than the crystalline phase. Sample 8 has a lower value of internal stresses due to the higher laser power used for its manufacturing and demonstrates no changing magnetic properties after annealing. Therefore, the heat treatment mode recommended for amorphous ribbons of this material [21] should be reevaluated for selective laser melting samples.

Further research requires the use of scanning strategies with different patterns and multiplicity, substrate heating and cooling experiments, and better optimization of physical and geometric process parameters and reaching more amorphous phase content.

## 4. Conclusions

The effect of selective laser melting parameters on microstructure and magnetic properties of 1CP alloy was investigated in this work.

Increasing the laser power leads to the enlargement of the array elements by melting larger amounts of powder. Increasing the offset parameter of the scanning strategy elements leads to a reduction in the density of the array by moving them farther apart. The hatch distance affected the structural unity of the elements in the pair and, consequently, the density of the array: an increase in this parameter results in a lack of fusion between the elements.Investigation of separate elements of samples 4 and 8 showed no differences in microstructure and phase composition characterized by the presence of solid solution α-Fe(Si), iron boride Fe_2_B and, probably, ordered solid solution Fe_3_Si. However, sample 8 obtained at *P* = 120 W is characterized by the layer shape expressed by the presence of arc sections on the boundary line, in contrast to sample 4 whose layer boundaries are expressed by smoother curves. Such influence of laser power on interlayers geometry is caused by deeper penetration of laser irradiation and non-uniform distribution of power across a spot of the laser beam.The DSC investigation of samples showed a practically complete absence of crystallization processes in sample 8 in the temperature interval of 400–600 °C in which visible crystallization peaks of initial powder. However, the primary heating curve of sample 4 is characterized by small peaks that indicate the presence of a small amount of amorphous phase in sample 4.The microhardness test results demonstrate no influence of laser beam power on samples manufactured with different laser power. The heat treatment contributes to a decreasing value of microhardness of the samples by about 40%.The study of magnetic properties showed insignificant differences in coercivity and close values of saturation magnetization for samples selective laser melted with different parameters. However, there are differences in hysteresis loop shape and values of residual magnetization: sample 4 has residual magnetization of 9.7 ± 0.2 emu/g, and sample 8 has residual magnetization of 10.8 ± 0.2 emu/g.

## Figures and Tables

**Figure 1 materials-15-04121-f001:**
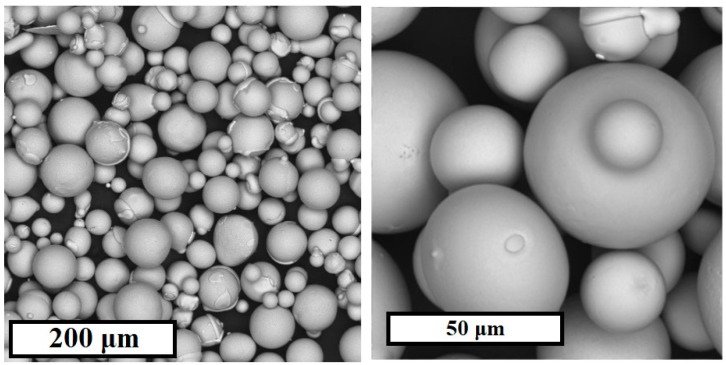
SEM image of 1CP powder.

**Figure 2 materials-15-04121-f002:**
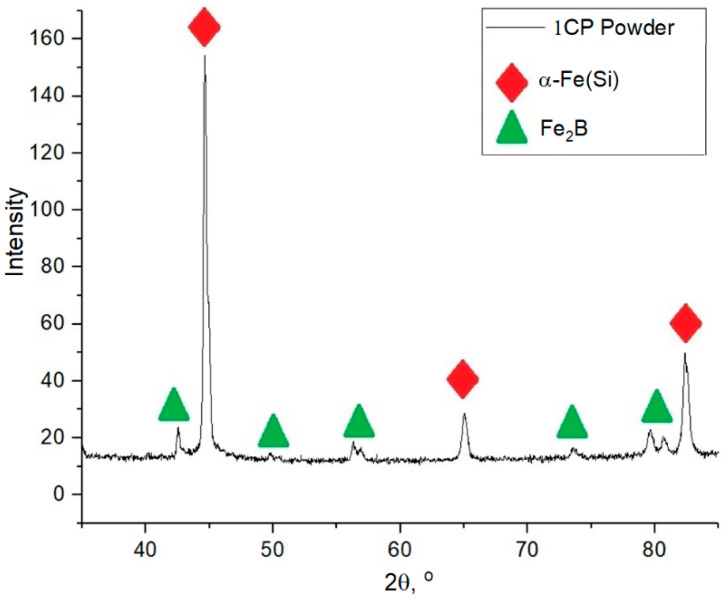
X-ray diffraction pattern of 1CP powder.

**Figure 3 materials-15-04121-f003:**
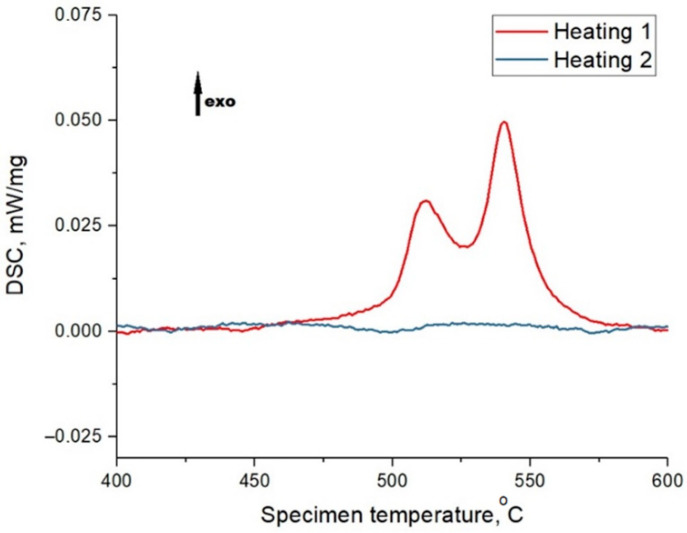
DSC heating curves for the 1CP powder (“exo” means that presented peaks are corresponded to exothermic process).

**Figure 4 materials-15-04121-f004:**
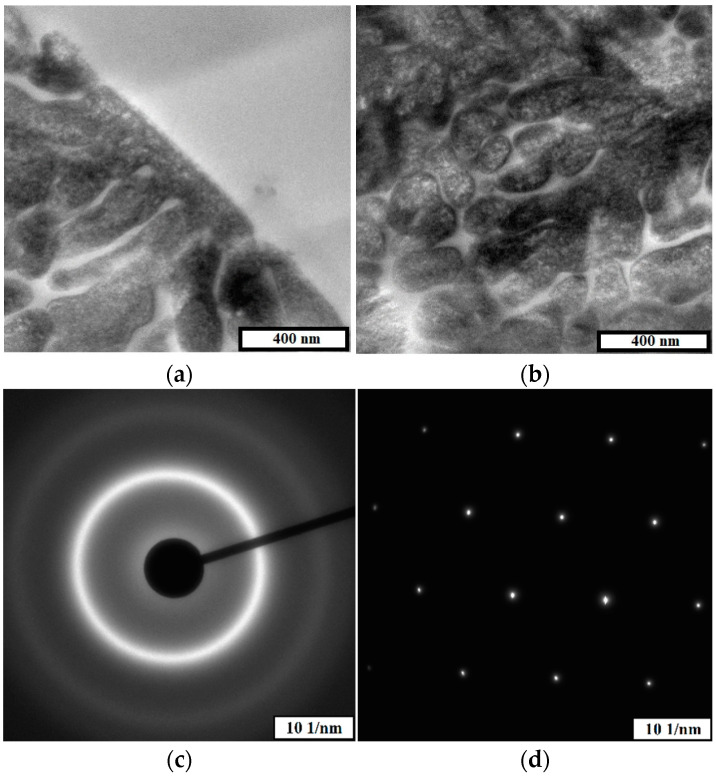
High-resolution TEM images of 1CP metal powder particles (**a**,**b**), electron diffraction pattern of the amorphous phase referring to the light region (**c**), electron diffraction pattern of the crystalline phase referring to the dark region (**d**).

**Figure 5 materials-15-04121-f005:**
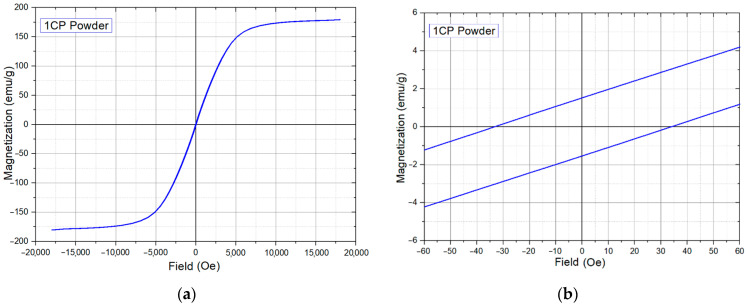
Magnetic properties measurement results of the 1CP powder: (**a**)—general view of hysteresis loop; (**b**)—enlarged area for coercivity estimation.

**Figure 6 materials-15-04121-f006:**
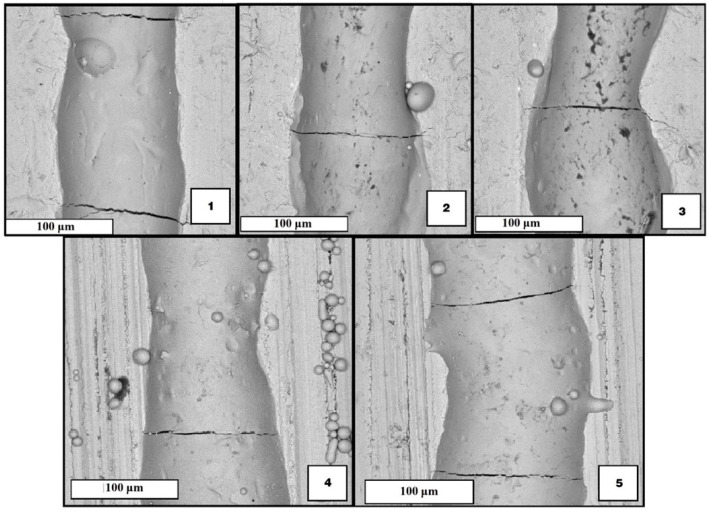
SEM images of top-views single tracks (numbers **1**–**5** denotes the building mode of Table 5).

**Figure 7 materials-15-04121-f007:**
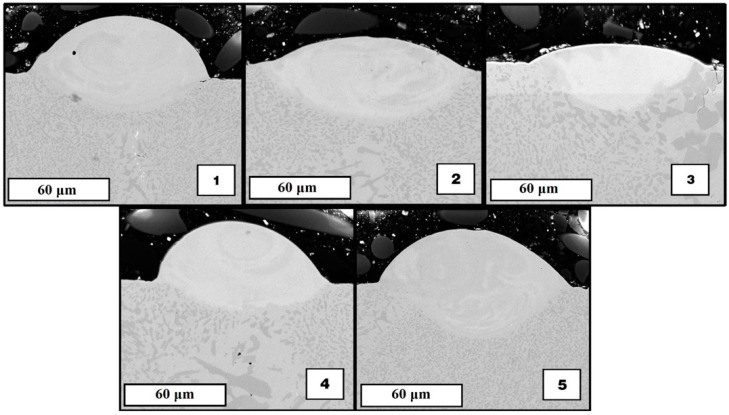
SEM images of cross-sectional views single tracks (numbers **1**–**5** denotes the building mode of Table 5).

**Figure 8 materials-15-04121-f008:**
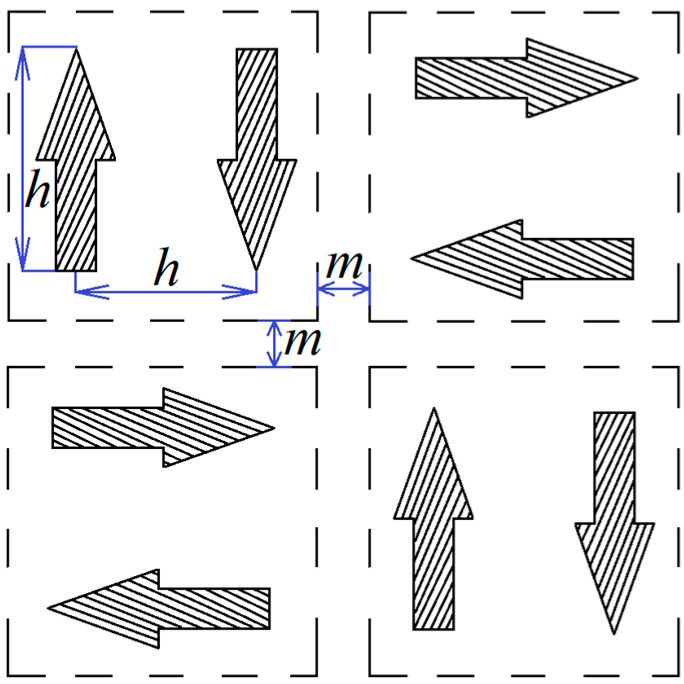
Sample building scheme (arrows in the cells indicates the scanning direction).

**Figure 9 materials-15-04121-f009:**
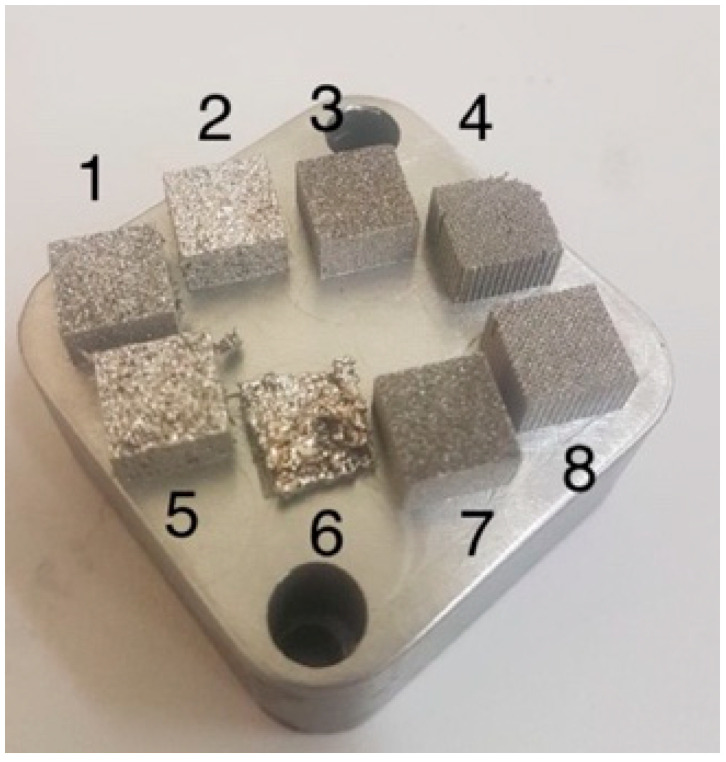
Image of samples manufactured by the SLM process from 1CP alloy powder (numbers **1**–**8** denotes the building mode of Table 7).

**Figure 10 materials-15-04121-f010:**
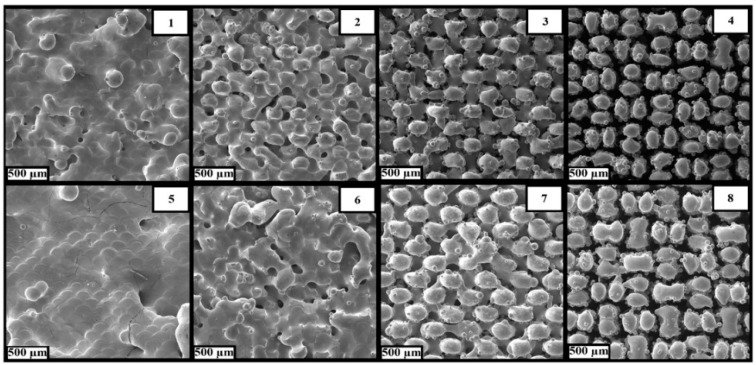
SEM images of samples (top view) produced by selective laser melting with parameters according to Table 6 (numbers **1**–**8** denotes the building mode of Table 7).

**Figure 11 materials-15-04121-f011:**
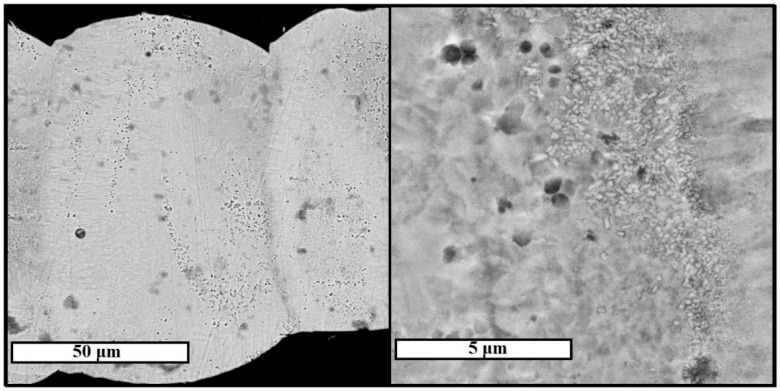
Microstructure of columnar element of sample 4, studied in backscattered electrons mode of SEM.

**Figure 12 materials-15-04121-f012:**
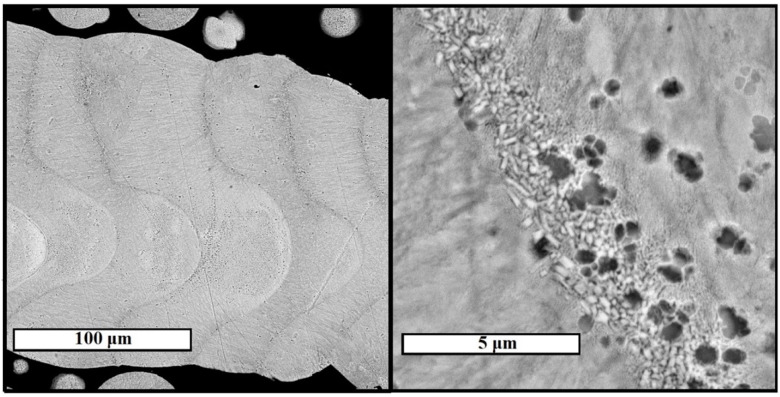
Microstructure of columnar element of sample 8, studied in backscattered electrons mode of SEM.

**Figure 13 materials-15-04121-f013:**
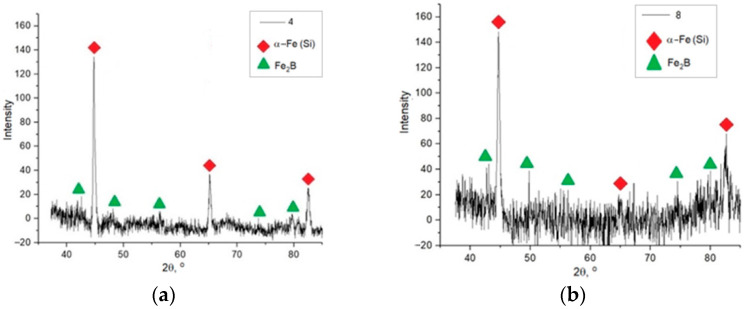
X-ray diffraction patterns of samples 4 (**a**) and 8 (**b**).

**Figure 14 materials-15-04121-f014:**
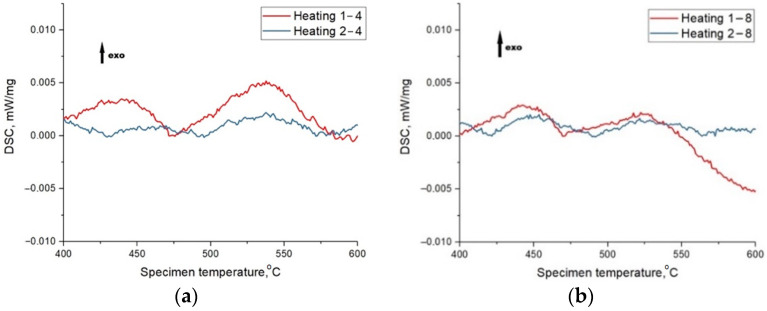
DSC curves of samples 4 (**a**) and 8 (**b**) (“exo” means that presented peaks are corresponded to exothermic process).

**Figure 15 materials-15-04121-f015:**
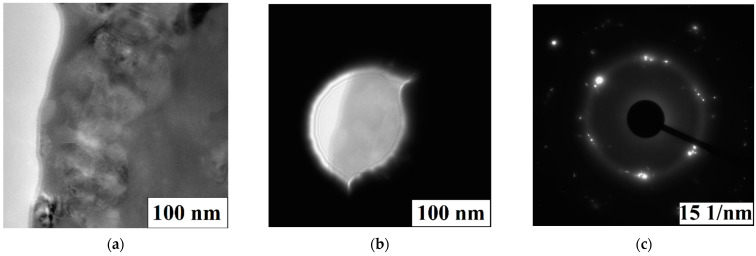
High resolution TEM images of sample 4 (**a**), electron diffraction pattern (**c**) of referring to the region (**b**).

**Figure 16 materials-15-04121-f016:**
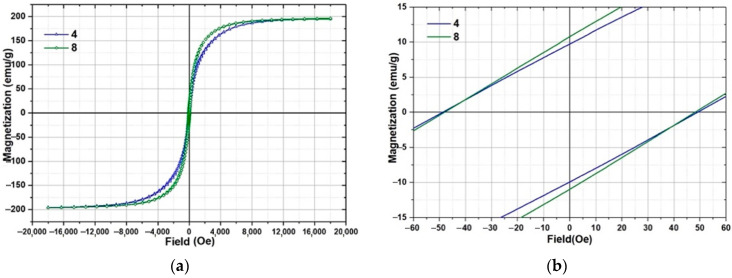
Magnetic properties measurements results of samples 4 and 8: (**a**)—general view of hysteresis loop; (**b**)—enlarged area for coercivity estimation.

**Figure 17 materials-15-04121-f017:**
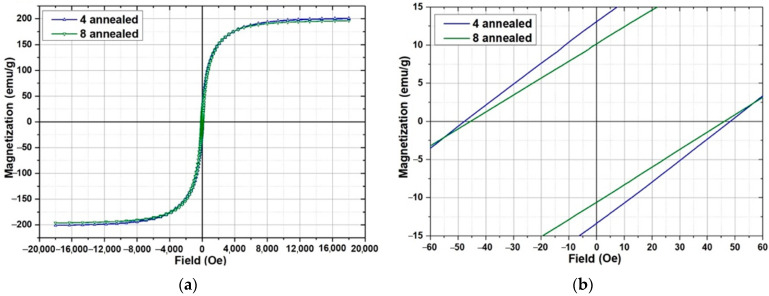
Magnetic properties measurements results of samples 4 and 8 after annealing heat treatment: (**a**)—general view of hysteresis loop; (**b**)—enlarged area for coercivity estimation.

**Table 1 materials-15-04121-t001:** The chemical composition of 1CP powder.

Fe, %	C, %	Si, %	B, %	O, %
Bal.	0.05	2.38	6.42	0.03

**Table 2 materials-15-04121-t002:** Particle size distribution of 1CP powder.

*d*_10_, μm	*d*_50_, μm	*d*_90_, μm
13.6	41.8	75.3

**Table 3 materials-15-04121-t003:** Physical and technological properties of the 1CP powder.

Flow Rate, s/50 g	Apparent Density, g/cm^3^	Skeletal Density, g/cm^3^
13	4.21	7.41

**Table 4 materials-15-04121-t004:** Magnetic properties of the 1CP powder.

Coercivity, Oe	Saturation Magnetization emu/g	Residual Magnetization emu/g	Rectangularity Factor
33 ± 1	180 ± 3	1.55 ± 0.2	0.008

**Table 5 materials-15-04121-t005:** Single track modes.

Sample Number	*P*, W	*V*, mm/s	Linear Energy Density, J/m
1	60	800	75
2	60	1000	60
3	60	1200	50
4	90	1200	75
5	120	1200	100

**Table 6 materials-15-04121-t006:** Geometric characteristics of the tracks.

Sample	Width, µm	Fusion Depth, μm	Deposit Height, μm
1	108.7	23.1	33.8
2	126.4	33.4	16.8
3	105.6	28.2	12.7
4	105.7	20.6	25.6
5	113.1	31.2	33.1

**Table 7 materials-15-04121-t007:** Modes of selecting laser melting used for manufacturing samples.

Sample	*P*, W	*h*, μm	*m*, μm
1	90	100	50
2	90	100	0
3	90	100	100
4	90	200	200
5	120	100	50
6	120	100	0
7	120	100	100
8	120	200	200

**Table 8 materials-15-04121-t008:** Magnetic properties of the samples.

Sample	Coercivity, Oe	Saturation Magnetization, emu/g	Residual Magnetization, emu/g	Rectangularity Factor
4	49 ± 1	195 ± 3	9.7 ± 0.2	0.050
8	48 ± 1	195 ± 3	10.8 ± 0.2	0.055
4 annealed	48 ± 1	200 ± 3	13.1 ± 0.2	0.065
8 annealed	46 ± 1	195 ± 3	10.2 ± 0.2	0.052

**Table 9 materials-15-04121-t009:** Hardness of the samples obtained (HV_0.3_).

Sample	Point 1	Point 2	Point 3	Point 4	Point 5	Mean
4	2226	1998	1986	1841	1854	1981
8	1785	1895	1883	1932	1707	1840
4 annealed	1108	1063	1254	1315	1402	1228
8 annealed	1118	1112	1149	1175	1212	1153

## Data Availability

Not applicable.

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
