# Peer review of "Structure, Mechanical and Magnetic Properties of Selective Laser Melted Fe-Si-B Alloy"

_materials, 2022, doi:10.3390/ma15124121_

Round 1

Reviewer 1 Report

This paper investigated the structure, mechanical and magnetic properties of selective laser 2 melted Fe–Si–B alloy. It is meaningful to investigate the effect of thermal treatment of samples on their magnetic and mechanical properties. This paper is recommended to be published. However, there are some questions which should be fixed.

Point 1: Lines 11 SLM should be given the full name. The first occurrence of the abbreviation requires the full name.

Point 2: In "Materials and Methods section", how many tests were repeated should be given.

Point 3: Please explain the particle size distribution of 1CP powder in table 1.

Point 4: Please pay attention to the subscripts, decimal point, and unit and in table 2.

Point 5: Give the unit of 2θ in Figure 2, and all charts should be standardized, the format should be consistent , and the icons should be put in suitable position.

Point 6: Lines 127, Figure 3 shows the results of a DSC investigation of the powder material. should put it together with the next paragraph. All the similar problems should be revised in this paper.

Point 7: Lines 157-164, please give the detailed description of magnetic properties, not just the figure and table.

Point 8: Physical quantities should be in italics.

Author Response

Thank you for your time for reviewing the manuscript and valuable comments.

Point 1: Lines 11 SLM should be given the full name. The first occurrence of the abbreviation requires the full name.

Answer: Thank you for your remark. Correction with full name of “selective laser melting” in abstract has been made. We hope this increase readability of the manuscript.

Corrected text: «Selective laser melting parameters were investigated through the track study, more suitable ones were found: laser power P = 90, 120 W; scanning speed V = 1200 mm/s».

 Point 2: In "Materials and Methods section", how many tests were repeated should be given.

Answer: The following text added to the manuscript:

In the magnetic measurements paragraph: “Magnetic measurements were carried out on 2 sets for each type of samples.

In the hardness paragraph: “To determine the mean value, 5 tests were performed.”

 Point 3: Please explain the particle size distribution of 1CP powder in table 1.

Answer: The following text was added to the manuscript: ” Initial powder particle size is Gaussian distributed with mean value of 41,8 μm.”.

 Point 4: Please pay attention to the subscripts, decimal point, and unit and in table 2.

Answer: Thank you, we have made correction of stiles of data in table 2.

 Point 5: Give the unit of 2θ in Figure 2, and all charts should be standardized, the format should be consistent , and the icons should be put in suitable position.

Answer: The unit has been added in Fig.2 and 13, all charts were checked and corrected.

 Point 6: Lines 127, Figure 3 shows the results of a DSC investigation of the powder material. should put it together with the next paragraph. All the similar problems should be revised in this paper.

Answer: The data presentation has been revised throughout the manuscript.

 Point 7: Lines 157-164, please give the detailed description of magnetic properties, not just the figure and table.

Answer: The following text added: “1CP can be considered as a soft magnet with a relatively high coercive force and a low residual magnetization, but a huge saturation magnetization”.

 Point 8: Physical quantities should be in italics.

Answer: All of the physical quantities was made in italics.

Reviewer 2 Report

The authors studied the effects of SLM process parameters on the magnetic and mechanical properties of samples. Overall, this work is presented well and this manuscript is very concise and clear. However, there are two concerns before being published.  The authors considered laser power, hatch distance, and offset m. However, another factor of scanning speed is not considered. The author should address this concern. Another concern is the laser power effect on samples manufactured. No influence should be elaborated more.   

Author Response

Thank you for your time for reviewing the manuscript and valuable comments.

During single track study we have made preliminary tests using different values of laser power P and scanning speed V, after is it has been chosen those ones which provided continuous tracks, their investigation is presented in the manuscript.

The following text has been added to revised version of the manuscript:

In order to determine the range of applicability parameters for the selective laser melting process, a series of single tracks were melted on a 1CP substrate using different values of laser power P and scanning speed V, which were selected after the preliminary tests have been made with various values of laser power and scanning speed and provided continuous tracks.

In the "Selective laser melting of samples investigation" section scanning speed was fixes for all of SLMed samples (the value was 1200 mm/s, lines 357-358 of revised version of manuscript) to investigate the laser power and other parameters effects. Laser power effect is minor in the investigated range of values but marked in qualitative effect that lies in the presence of amorphous phase, which is presented in the paper.

Reviewer 3 Report

The paper “Structure, mechanical and magnetic properties of selective laser melted Fe-Si-B alloy” by Erutin and co-authors reports on a parameter study of additive manufacturing of a partially amorphous Fe-based precursor material. The authors studied the powder properties and performed a variation of the printing parameters in single tracks and by creating rectangular samples. Additive manufacturing is currently a highly investigated topic and new material compositions and alloys are desperately sought for by research and industry

This said unfortunately in the current state the publication has to be rejected for resubmission as for the reader it is not consistently possible to understand the novelty, the aim and the main outcome of the present study. If this has been resolved the work might, in the opinion of the reviewer, apply for resubmission in the journal.

Some points that lead to this reasoning are given below.

  • The authors claim that they want to thoroughly understand the mechanisms during additive manufacturing. The present paper rather reports a collection of experimental observations. It is not linking the structural data to the magnetic and mechanic properties. To some extent the printing parameters are linked to the experimental data.
  • In the methods section the abbreviation of the alloy 1CP is given. One can find a Russian data sheet on this material but the exact composition of the powder has not been given and remains elusive to the reader. The reviewer attempted to find some data in other languages but this data might not be accessible any more from outside Russia after the war in Ukraine was started by Russia in February of this year. Please provide the composition of the alloy.
  • Does the composition of the powder diverge from the nominal composition?
  • What is the oxygen content of the powder?
  • The description of the methods is extremely vague particularly with respect to the printing processes. Was the type of laser and wavelength given? Optical configuration? Gas flow…?
  • Positively the quality of the TEM data has to be highlighted. Was TEM also performed on the printed samples. Especially sample e should show some amorphous material if the claims of the authors are correct. Please provide TEM data to prove the amorphous phases in sample 4.
  • Single track method is an acceptable approach to optimize processing parameters, although it would be better to mention how it is decided to select the 5 regimes for the power or scanning speed.
  • The Mechanism of the cracking need to be clearly proved in the line 178-180 for both low and high energy density or the claim has to be removed.
  • lines 202-206 are unclear, it is better to use a schematic to better understanding of the designing, “m” and “h” parameters.
  • The impact of presence or absence of amorphous phases on the properties of the printed parts is unclear.
  • The purpose of doing heat treatment is unclear (magnetic?)
  • The impact of amorphous phases in the microstructure comparing to the heat treatment need to be mentioned.
  • Some of the questions answered in the conclusion part which must be explained in the discussion part, as well.
  • What is the role of the grain/crystal size? How does it affect magnetic properties?
  • For the DSC data onset temperatures and transformation enthalpies should be provided.

To conclude the paper needs substantial revision before it may be considered for publication. The topic itself has some value and the collected experimental data is quite broad. The discussion of the paper is however very week as the linkage of the data to the produced structures is missing. The English is good.

Author Response

Thank you for your time for reviewing the manuscript and valuable comments.

Please find responses on comments in attached file.

Reviewer 4 Report

The Authors studied the original 1CP powder and it was founded that powder material partially consists of amorphous phase, which crystallization begins at 450°C and ends at 575°C, also they succeeded to investigate the SLM parameters through the track study.
The results showed from the X–Ray diffraction of samples presence of –Fe(Si) and Fe2B, in addition the SEM–image analysis shows presence of ordered Fe3Si in both samples.

Also, the intensively experimental work done by the authors in the presences of the annealed samples show 40% less microhardness; annealed sample containing amorphous phase shows higher soft–magnetic properties: 2,5% higher saturation magnetization, 35% higher residual magnetization and 30% higher rectangularity coefficient.

In general, the work done is showing high novelty in illustrating and presenting up the results.

Author Response

Thank you for your time and positive estimation of our work!

Round 2

Reviewer 3 Report

The questions raised by the reviewer have been answered by the authors and the paper does now provide a very useful contribution to the community that can also be reproduced. The TEM data has been completed an is now a highlight of the present work.

I do recommend the paper for publication!